Computational approach for counting of SISH amplification signals for HER2 status assessment

Rehman Zaka Ur 1
Ahmad Fauzi Mohammad Faizal faizal1@mmu.edu.my 1
Wan Ahmad Wan Siti Halimatul Munirah 1 2
Abas Fazly Salleh 3
Cheah Phaik Leng 4
Chiew Seow Fan 4
Looi Lai-Meng 4
1 Faculty of Engineering, Multimedia University , Cyberjaya , Selangor , Malaysia
2 Institute for Research, Development and Innovation, International Medical University , Bukit Jalil, Kuala Lumpur , Malaysia
3 Faculty of Engineering and Technology, Multimedia University , Ayer Keroh, Malacca , Malaysia
4 Department of Pathology, University Malaya Medical Center , Kuala Lumpur , Malaysia
Wan Shibiao
Electronic publication date: 2024 Oct 23
Publication date: 2024
Volume: 10
Electronic Location ID: e2373
Received 2024 Jul 5; Accepted 2024 Sep 9
Copyright: ©2024 Rehman et al.
Copyright year: 2024
Copyright holder: Rehman et al.
License: This is an open access article distributed under the terms of the Creative Commons Attribution License, which permits unrestricted use, distribution, reproduction and adaptation in any medium and for any purpose provided that it is properly attributed. For attribution, the original author(s), title, publication source (PeerJ Computer Science) and either DOI or URL of the article must be cited.
License URL: https://creativecommons.org/licenses/by/4.0/

Keywords: Biomarkers, Deep learning, Digital pathology, Human epidermal growth factor receptor 2, Silver in situ hybridization (HER2-SISH)

Funding: Fundamental Research Grant Scheme (FRGS), Malaysia FRGS/1/2020/ICT02/MMU/02/10 This work was supported by the Fundamental Research Grant Scheme (FRGS), Malaysia (FRGS/1/2020/ICT02/MMU/02/10). The funders had no role in study design, data collection and analysis, decision to publish, or preparation of the manuscript.

==============================
The human epidermal growth factor receptor 2 (HER2) gene is a critical biomarker for determining amplification status and targeting clinical therapies in breast cancer treatment. This study introduces a computer-aided method that automatically measures and scores HER2 gene status from invasive tissue regions of breast cancer using whole slide images (WSI) through silver in situ hybridization (SISH) staining. Image processing and deep learning techniques are employed to isolate untruncated and non-overlapping single nuclei from cancer regions. The Stardist deep learning model is fine-tuned on our HER2-SISH data to identify nuclei regions, followed by post-processing based on identified HER2 and CEP17 signals. Conventional thresholding techniques are used to segment HER2 and CEP17 signals. HER2 amplification status is determined by calculating the HER2-to-CEP17 signal ratio, in accordance with ASCO/CAP 2018 standards. The proposed method significantly reduces the effort and time required for quantification. Experimental results demonstrate a 0.91% correlation coefficient between pathologists manual enumeration and the proposed automatic SISH quantification approach. A one-sided paired t-test confirmed that the differences between the outcomes of the proposed method and the reference standard are statistically insignificant, with p-values exceeding 0.05. This study illustrates how deep learning can effectively automate HER2 status determination, demonstrating improvements over current manual methods and offering a robust, reproducible alternative for clinical practice.

Introduction

Breast cancer remains a leading cause of cancer-related mortality among women worldwide, underscoring the critical need for accurate diagnostic and therapeutic strategies. Human epidermal growth factor receptor 2 (HER2) gene amplification, observed in 15–20% of breast cancers, is associated with aggressive tumor behavior and poorer prognosis (Jemal et al., 2011; Dj, 1987). Determining HER2 status is crucial for selecting appropriate treatment strategies, guiding the use of HER2-targeted therapies such as trastuzumab.

Current diagnostic methods, including fluorescence in situ hybridization (FISH) and immunohistochemistry (IHC), suffer from methodological inconsistencies and subjective interpretation (Kang et al., 2009). Variations in tissue preparation, staining protocols, and inter-observer variability can lead to discordant results, affecting clinical decision-making (Rüschoff et al., 2010). Cancerate details of different stains importance and there purpose are analysed in Fig. 1 and clearly analysed the importance of SISH for HER2 scoring.

Figure 1 A concrete visual analysis of preferring the ISH stain our IHC stains is given in the picture.

Silver in situ hybridization (SISH) has emerged as a promising alternative to FISH, offering several advantages. SISH allows for the visualization of gene amplification using bright-field microscopy, which is more familiar to pathologists and less reliant on specialized equipment (Gutierrez & Schiff, 2011). Additionally, SISH provides a more permanent record of the results as the slides can be stored and re-evaluated if necessary, unlike FISH slides which fade over time. SISH also reduces issues related to fluorescence signal degradation and background autofluorescence, which can complicate FISH analysis.

Despite these advancements, accurate and reproducible HER2 scoring from SISH images remains challenging. Manual assessment by pathologists is time-consuming and susceptible to human error, particularly in cases with ambiguous signal patterns or tissue artifacts. Figure 2 shows a sample SISH regions of interest with detailed description of information of HER2 and CEP17 signals and considerable nuclei characteristics.

Figure 2 Visual representation of HER2-SISH image with detailed description of nuclei and predictive biomarker.

The integration of image analysis and machine learning presents a transformative opportunity in automating HER2 status determination from SISH images. Deep learning techniques, particularly convolutional neural networks (CNNs), have shown promise in automating pathological image analysis tasks, improving accuracy and reducing variability (LeCun, Bengio & Hinton, 2015).

This study aims to address these challenges by proposing a novel method for automated HER2 scoring from SISH images, thereby demonstrating the application of artificial intelligence (AI) techniques to a critical medical problem. The key contributions of this work include:

1. Introducing a robust methodology for automated detection and scoring of HER2/CEP17 biomarkers from identified nuclei in SISH stain images, addressing the practical need for accurate and reproducible HER2 status assessment.

2. Evaluating the concordance between computer-aided HER2 status determination and expert pathologist assessments using SISH, thus demonstrating the improvement of the proposed method over current practices.

3. Investigating factors influencing inter-observer interpretative reproducibility among pathologists, providing insights into the practical limitations and potential of the technique.

The structure of this paper is organized as follows: ‘Material and Methodology’ details the proposed image analysis and machine learning techniques for HER2 scoring, including the specific deep learning models and algorithms employed. ‘Experimental Setup’ presents experimental results, comparing the performance of the proposed method with existing methodologies to demonstrate its effectiveness and practical utility. Finally, ‘Conclusion’ summarizes findings, discusses implications for clinical practice, and outlines directions for future research, ensuring the article is technically robust and clearly presented.

Background Study

Numerous traditional methodologies for the automated extraction of features from microscopic images have been developed, including techniques for spot detection and counting (Bright & Steel, 1987; Olivo-Marin, 2002). However, a notable upsurge in the creation and effective execution of deep learning-based applications for the categorization of pathological microscopic images has occurred recently (LeCun, Bengio & Hinton, 2015). Typically, these image classification endeavors employ CNNs, which utilize a series of convolutional layers and non-linear transformations to process input data, facilitating the generation of high-level abstractions for classification purposes (Krizhevsky, Sutskever & Hinton, 2012).

Deep learning, particularly CNNs, has revolutionized the field of pathology by automating image analysis tasks with high accuracy. CNNs have been widely used for image classification, object detection, and segmentation tasks. In pathology, these models have shown significant promise in automating the analysis of histopathological images, detecting tumors, and classifying cancer types (Olivo-Marin, 2002; Xu et al., 2014; LeCun, Bengio & Hinton, 2015; Abbas et al., 2023).

In the context of HER2 quantification, several studies have leveraged deep learning techniques to improve the accuracy and efficiency of analysis. For instance, CNNs have been applied to the identification and counting of fluorescent signals inside nuclei in FISH images (Gudla et al., 2017), and to chromosome segmentation in multicolor FISH images (Pardo, Morgado & Malpica, 2018). SpotLearn, a supervised machine learning system, has been utilized for the precise identification of FISH signals from fluorescence microscopy channels (Pezoa et al., 2016). These advancements highlight the potential of deep learning in automating HER2 status determination, reducing the reliance on manual counting by pathologists.

Recent studies have further demonstrated the capabilities of deep learning in pathology. For example, Masmoudi et al. (2009) proposed a three-phase algorithm involving membrane staining classification, cell nuclei segmentation, and color pixel classification to distinguish between overexpressed and underexpressed HER2 regions. Similarly, Saha & Chakraborty (2018) introduced Her2Net, a deep learning model using a trapezoidal long short-term memory (LSTM) connection topology for segmenting cell membranes and detecting nuclei. These studies underscore the importance of incorporating advanced deep learning models to enhance the accuracy and reliability of HER2 quantification.

Numerous publications detail approaches for automated HER2 amplification status assessment using FISH, CISH, and DISH stains based on HER2/CEP17 signals, alongside various other applications. Most of these studies utilize a combination of classical image processing and deep learning techniques to detect signals (Höfener et al., 2019; Hossain et al., 2019; Wang et al., 2023). For instance, Wang et al. (2023) proposed a weakly supervised Cascade R-CNN(W-CRCNN) model to automatically detect HER2 overexpression in HER2 DISH and FISH images from clinical breast cancer samples. Hossain et al. (2019) selected untruncated and non-overlapping singular nuclei from cancer regions using color unmixing and machine learning techniques, then detected and counted HER2 and chromosome enumeration probe 17 (CEP17) signals based on RGB intensity per nucleus. Similarly, Höfener et al. (2019) proposed a novel density-based approach to quantify FISH signals, which quantifies signals through the integral over a density map predicted by deep learning instead of detecting individual signals.

Despite these advancements in automated methods for various stains, each has its limitations and issues, as noted in the literature. Our work is pioneering in the application of automated HER2 status assessment using SISH stain, an alternative to the aforementioned stains. However, challenges remain in achieving accurate and reproducible assessment of HER2 status with SISH images. Variations in tissue preparation, signal intensity, and overlapping cells can affect the accuracy of automated methods. Future research should focus on improving the robustness of deep learning models to handle these variations, as well as integrating multimodal data to enhance diagnostic accuracy (Wang et al., 2022). Additionally, there is a need for large, annotated datasets to train and validate these models effectively. There is no current work on SISH images specifically, so the background study is focused on different stains and methodologies relevant to our proposed approach.

Material and Methodology

Material

We obtained 50 whole slide images (WSIs) of HER2-SISH stain from our collaborating institution, the University Malaya Medical Center (UMMC). These WSIs were scanned at a magnification of 40 × using the 3DHistech Pannoramic DESK scanner, with image dimensions typically ranging from 80,000 to 200,000 pixels. From each WSI, our pathologist extracted five regions of interest (ROIs), resulting in a total of 250 ROIs. These ROIs varied in size, from approximately 859 × 755 to 5,451 × 3,136 pixels per ROI. For our experiments, we selected 70 ROIs containing a minimum of 20 nuclei with clearly defined boundaries for analysis.

To establish a gold standard, nuclei within the selected regions across all 70 ROIs were manually annotated and subsequently validated by an expert pathologist. Given the impracticality of manually segmenting all nuclei within each region, we employed a semi-automated approach for annotation. A senior pathologist then reviewed and validated the annotated nuclei according to clinical practice guidelines. This study was approved by the University of Malaya Medical Centre’s Medical Research Ethics Committee (UMMC-MREC), Ref MREC-ID 202195-10555. Table 1 provides detailed statistical information regarding the images.

Table 1 Quantitative details of WSI and region-based annotations by experts.

Sr. No.	Description	Amplified	Non amplified	Normal	
1	Number of WSI	25	21	0	
3	Number of ROI marked by experts	145	100	399	

System overview

The proposed method uses pathologist annotations of invasive tumor locations on digital SISH slides for automated quantification, and it works with high-resolution SISH WSIs (e.g., 0.13 µm/pixel). To distinguish between HER2 and CEP17 signals, a high-resolution image is required. The proposed method for determining the status of HER2 amplification is shown in Fig. 3. An automatic sectioning machine is used to first section tissue blocks containing tumors, creating 4 µm serial sections for SISH staining. These sections are then stained using an automatic staining machine to prepare SISH glass slides. Subsequently, the SISH glass slides are scanned using a WSI scanner at 40 × magnification to generate digitized WSIs. Two specialized breast pathologists annotate the invasive cancer ROIs on the SISH WSIs.

Figure 3 The overall concept and workflow of the proposed system, from slide preparation to the CAD scoring system.

Some component images used in this figure are sourced from https://www.freepik.com/. Specifically, the paraffin wax biopsy sample tissue silhouette is adapted from https://www.freepik.com/premium-photo/laboratory-assistant-works-paraffin-wax-dispenser-tissue-embedding-machine_21821215.htm, and the digital slide scanner from https://www.freepik.com/free-vector/smart-industry-icon_23182662.htm. Copyright ©2010-2024 Freepik Company S.L., used under a free license. All other components are our own setup, including the glass slides from our dataset.

We extracted the annotations from the HER2-SISH WSI. The proposed technique used have to follow and extract the individual nuclei, HER2 signals (black dots), and CEP17 signals (Red dots) from the ROI. The signals are then listed for each nuclei, with only those where CEP17 ≥ 2 taken into account. A minimum of 20 nuclei with the greatest HER2-CEP17 differentiation values are chosen by the approach. The overall CEP17 and HER2 signal counts are then calculated for the chosen nuclei. Next, the HER2 amplification status is classified as either “positive” or “negative” according on the 2018 ASCO/CAP standards (Xu et al., 2019).

The model makes sure that another set of at least 20 nuclei is measured in a similar manner if the number of HER2 copies per nucleus falls into a HER2 status category that necessitates counting an extra 20 nuclei. The average number of HER2 copies per cell from a minimum of 40 nuclei and the computed HER2 to CEP17 ratio are then used to establish the amplification status. The clinical practise for the computation of amplification status is shown in Fig. 4A, Our proposed flowchart for computation of HER2 scoring is shown in Fig. 4B, and Fig. 4C shows the nuclei detection workflow.

Figure 4 (A) The diagram illustrating HER2 scoring proposed in the clinical practise. (B) The approach used in our paper. The approach for the nuclei segmentation workflow.

Preprocessing

Histopathological images are notably scanned at high magnification level, due to that the image size is large and complex to handle, presenting many challenges for computer based algorithms as compared to normal images. The rich texture information contained within these high-resolution images is crucial for successful cancer diagnosis across various color contrast differences (Shi et al., 2016). To address the challenges posed by the size and contract difference complexity of these images, we implemented a basic image normalization technique aimed at standardizing the contrast and color range of the images. Given an array x comprising elements x1, x2, …, xn, the normalization process scales each element xi to a new value xi′ such that all new values fall within a specified range [lower_bound, upper_bound]. The formula for this normalization is as follows: (1) xi′=xi− minxmaxx−minx×upper_bound−lower_bound+lower_bound

In this formula:

• xi′ is the normalized value of xi,

• min(x) is the minimum value in array x,

• max(x) is the maximum value in array x,

• lower_bound is the lower bound of the desired range,

• upper_bound is the upper bound of the desired range.

This normalization procedure ensures that the values in x are linearly scaled to fit the predefined range [lower_bound, upper_bound], with this specific range set to [1, 99.8] for our study. This approach facilitates the processing and analysis of histopathological images by helping to mitigate variations in lighting and staining, thereby improving the consistency and reliability of subsequent analyses.

HER2 and CEP17 signal detection

In SISH images, HER2 signals appear as black dots, with intensity ranging from very dark to bright black. These signals can vary in size both within and across slides, reflecting the biological variability of the signal within nuclei. HER2 probes may also cluster together, with cluster sizes also varying. CEP17 signals, on the other hand, appear as red dots, with their intensity also ranging from dark to bright.

For signal detection, we are utilizing basic image processing techniques, we first remove the background of the image and separate the signals by color thresholding. The second step involves removing very faint signals that are likely noise. In third step, we segment the nuclei based on fine tuned stardist model that we train on our dataset, in the fourth step, apply scoring criteria that consider only those nuclei with at least two CEP17 signals for HER2 scoring. We then post-process the segmented nuclei image based on this signal criteria. Figure 5 is basically visualize the process of HER2 scoring have been purposed out in our method.

Figure 5 (A–C) Nuclei visualization of nuclei detection by using deep learning model. (B) Useless nuclei; (C) useful nuclei, and how much possible useless nuclei are removed automatically in the postprocessing part.

Nuclei detection

In our proposed method, the assessment relies on at least 20 nuclei selected from the tissue sample. Given the abundance of nuclei in the selected regions, it becomes crucial to discern those suitable for quantification. Nuclei containing overlapping or partially missing regions need to be excluded, along with false detections common in existing nuclei detection methods. Therefore, our nuclei detection method comprises two stages:

• Identification of nuclei with at least two CEP17 signals.

• Exclusion of nuclei with missing or excessively overlapped information.

For this purpose, we utilized the Stardist model, which was fine-tuned on our data for nuclei detection. The Stardist model was trained and validated on annotations provided by experts. The performance of the model across several test cases is outlined in Table 2, showing how many nuclei were detected from the image and how many had at least two HER2 signals out of the detected nuclei.

Table 2 Performance comparison of fine-tuned models on training and testing datasets, illustrating their classification results.

Sr. No.	Image ID	Detected nuclei	Nuclei with 2 CEP17 signals	
1	429485533	385	240	
2	429485565	344	209	
3	430537704	117	95	
4	430537413	122	104	
5	429493059	125	83	
6	429493014	114	81	
7	168136732	156	75	
8	168136787	165	60	
9	168138726	36	33	
10	168138679	66	57	
11	429487680	812	604	
12	429487703	776	594	
13	429487730	780	598	
14	169284724	175	117	
15	169284650	93	57	
16	168138400	159	130	
17	168138327	37	17	
18	168138509	85	52	
19	168139617	122	99	
20	168139685	83	75	

The Stardist model is a deep learning-based method specifically designed for segmenting and classifying star-convex shapes, such as cell nuclei, in microscopy images. This model was chosen for its ability to accurately delineate individual nuclei, even in densely packed regions, by representing each object as a star-convex polygon. Key features of the Stardist model include:

• Polygon representation: Each nucleus is represented as a polygon, making it easier to handle overlaps and partial detections.

• Robust training: The model was fine-tuned using a dataset annotated by experts, ensuring high accuracy in nuclei detection.

• Efficient processing: By decomposing large image patches into smaller segments (256 × 256), the model maintains high performance while reducing computational load.

Initially, we prepared the dataset by generating rough nuclei labels using image processing techniques and pre-trained Stardist models. These labels were further refined through expert validation. While the model successfully detected all nuclei, post-processing was necessary to filter out nuclei based on predefined conditions. We applied basic color thresholding to identify CEP17 and HER2 signals within the main image, retaining only those nuclei with signals. Additionally, we implemented a condition to discard nuclei with significant overlap (i.e., overlapping by more than 50% of their area).

To optimize network training efficiency, large patches of data were initially decomposed into smaller patches of size 256 × 256, aligning with the default setting of the Stardist model (Fazeli et al., 2020). The training dataset was augmented with synthetically generated data, contributing 650 additional patches of the same size. Consequently, the training dataset comprised 650 and 570 patches, each sized 256 × 256. For testing purposes, no preprocessing was required; the regions of interest (ROIs) were directly passed to the trained model. The train-validation split was set at 80:20%. To minimize predicted object probabilities, we employed standard binary cross-entropy loss. For polygon distances, we utilized mean absolute error loss, with weighting determined by corresponding ground truth object probabilities. In essence, pixel-wise errors were scaled based on object probabilities, with background pixels having zero object probability not contributing to the loss calculation. Additionally, predictions for pixels closer to the center of each object were prioritized to enhance model accuracy in capturing fine details and preserving object boundaries.

The performance of the nuclei detection model was not directly evaluated, due to unavailability of nuclei annotation, but at the end the overall scoring is computed by following ways.

1. Correlation coefficient: Measures the correlation between the automated nuclei count and manual enumeration by pathologists. A high correlation coefficient (0.91 in our case) indicates that the model’s predictions closely match expert assessments.

2. Statistical significance (p-value): a one-sided paired t-test was conducted to compare the differences between the outcomes of the proposed method and the reference standard. p-values exceeding 0.05 indicated no significant statistical difference, validating the reliability of our approach.

Overall, Stardist’s robustness, efficiency, and suitability for handling the specific challenges of histopathology images made it an ideal choice for our study.

Signal quantification

Our methodology for signal quantification aligns with the ASCO/CAP 2018 guidelines (Xu et al., 2019), focusing on the accurate counting of CEP17 and HER2 signals within individual nuclei. To achieve a representative evaluation of HER2 amplification, the process involves selecting nuclei that display at least two signals of both CEP17 and HER2. This selection criterion ensures that the quantification is based on the most relevant nuclei, aiming to include at least 20 nuclei per image for analysis. In cases exhibiting extremely high positivity where no nuclei meet the criteria of at least two CEP17 signals, high positivity indicates that most of the nuclei in the sample are likely to show strong HER2 signals, but they may not all display the usual criteria of having at least two CEP17 signals. Therefore, in such cases, the methodology adapts to include nuclei with only one CEP17 signal to ensure that the analysis can still proceed and reflect the flexibility used in manual assessments by pathologists.

The quantification process is systematic. Initially, nuclei are arranged based on their differentiation value between HER2 and CEP17 signals, prioritizing those with the most significant discrepancies. This sorting is crucial for identifying the best representatives for HER2 amplification estimation. Table 3 shows the standard criteria, based on that we have analysed the amplification status is considered. Subsequent steps involve calculating the HER2 to CEP17 ratio and the average HER2 copy number per quantified nucleus, utilizing the following formulas: (2) HER2 Ratio=Total HER2 SignalsTotal CEP17 Signals.

(3) Amplification Status=Total HER2 SignalsNumber of Quantified Nuclei.

Table 3 The table is on the assessment of cancer status based on standard criteria of scoring values (Milanezi, Carvalho & Schmitt, 2008).

Variables	Status	
HER2/CEP17 < 1.8	Negative	
1.8 =  < HER2/CEP17 <  = 2.2	Equivocal	
HER2/CEP17 > 2.0	Positive	

These calculations facilitate the determination of the HER2 amplification status by examining the ratio of HER2 signals per quantified nuclei. The process ensures a thorough and precise evaluation, closely mirroring the criteria set forth in the ASCO/CAP guidelines.

Experimental Setup

In this section, we detail the experimental setup utilized for the development and evaluation of the proposed HER2 scoring pipeline for HER2-SISH histopathology. We discuss the selection of parameters, evaluation measures, and the performance analysis.

Parameter setting

The selection of parameters is crucial for the success of parametric-based methods. Our approach begins by preprocessing data and annotations from medical institutions to train deep learning models for automatic nuclei region detection. We divide non-overlapping ROI into small-sized images or patches of 512 × 512 × 3 dimensions, using 80% of these patches to fine-tune the transfer learning model with pre-trained weights from “Imagenet” and reserving 20% for model testing. For nuclei segmentation, we use the StarDist model and fine-tune it on our data.

After detecting nuclei, parameter selection for signal detection is critical. The success of signal detection significantly affects the identification of diagnosable nuclei. As detailed in the signal detection section, we use a Gamma adjustment-based technique, where selecting Sigma and alpha parameters is essential. These values are manually selected based on standard criteria. We experiment with three layer configurations: (0.9, 1), (0.5, 1), and (0.5, 2).

Hardware specifications

This subsection outlines the specifications of the computer system utilized for the HER2-SISH histopathology diagnosis:

• Operating System: Windows 11 (64 bit)

• CPU: 13th Gen Intel(R) Core(TM) i7-13620H @ 2.40 GHz

• RAM: 32 GB

• Graphics Card RAM Size: 8 GB

The proposed histopathology system is implemented using Python programming language with CUDA 11.2, cuDNN 8.2, and OpenCV 3.0. Additionally, we integrate the DeepZoom generator (Goode et al., 2013) for handling WSI.

Evaluation criteria

In the context of HER2/CEP17 signal counts, the Bland-Altman plot (Mansournia et al., 2021) can be used to compare the manual signal counts by experts to the counts obtained using the proposed method. This comparison will help in evaluating:

• The overall agreement between the manual and automated methods.

• Any systematic bias in the automated method.

• The consistency and reliability of the automated method across different samples.

By using the Bland-Altman plot, we can visually assess how well the proposed method performs in comparison to manual counting and make necessary adjustments to improve the method’s accuracy and reliability.

Visual results

We used fined tuned StartDist model for nuclei segmentation. Stardist is very popular model for segmenting the nuclei from an image in histopathology and it is particularly designed for segmenting the nuclei from histopathology images, in our HER2-SISH images cases, we also fine tuned the model and it performed very well nuclei segmenation. Figure 6 presents a visual outputs for nuclei and HER2/CEP17 signals to show the visual effectiveness of our proposed method. Column 1 showcases three sample images extracted from WSIs that used in our analysis. These images serve as the baseline for comparison, highlighting the complexity and variability in the tissue samples. Column 2 images illustrate the performance of our trained StarDist model in effectively segmenting and overlaying the nuclei regions on the original samples images. Column 3 images are represented the combined segmentation of both nuclei and HER2/CEP17 signals. This dual-segmentation approach is essential for precise quantification and scoring of HER2 status. The images in this column demonstrate the proficiency of our method in detecting and segmenting the HER2/CEP17 signals, which are critical for accurate HER2 scoring.

Figure 6 Illustrative examples of HER2-SISH patch samples.

Column one are original patches, column 2 show the segmented nuclei and column 3 show the detection signals of the nuclei.

Quantification results

The evaluation of our proposed method was carried out through the quantification of HER2 status on the expert mark regions from 20 WSI cases, with a comparative analysis against manual SISH quantification conducted by experts. Table 4 presents the results of this comparison, focusing on HER2 scoring through both manual SISH and the proposed SISH quantification method across these cases, out of a total of 46 evaluated. Following the ASCO/CAP 2018 recommendations, our approach calculates the HER2/CEP17 ratio to assess the HER2 amplification status. According to the 2018 ASCO/CAP standards, HER2 amplification status is characterized as follows:

Table 4 Comparison of HER2 scoring through manual SISH and proposed SISH quantification method.

Sr. No.	Image ID	Detected nuclei	Nuclei with 2 CEP17 signals	HER2 signals	CEP17 signals	Ratio (HER2/CEP17)	Expert	Status	
Non amplified regions						
1	168136732	156	75	294	155	1.90	1.7	Neg	
2	168136787	165	60	217	126	1.72	1.7	Neg	
3	168138679	66	57	460	287	1.60	1.2	Neg	
4	168138726	36	33	219	150	1.46	1.2	Neg	
5	169282562	60	44	239	235	1.02	1.5	Neg	
6	169282603	204	140	895	815	1.10	1.5	Neg	
7	169282812	55	46	361	258	1.40	1.2	Neg	
8	169282871	29	24	254	170	1.49	1.2	Neg	
9	169283911	30	14	57	41	1.39	0.6	Neg	
10	169284039	39	23	75	29	2.59	0.6	Neg	
11	169284198	100	62	289	212	1.36	1.3	Neg	
12	169284222	78	50	191	150	1.27	1.3	Neg	
13	168136732	156	75	294	158	1.86	1.7	Neg	
14	168136787	165	60	228	126	1.81	1.7	Neg	
15	169285625	127	110	886	569	1.56	1.4	Neg	
16	169285655	118	103	920	601	1.53	1.4	Neg	
17	169285908	165	124	1225	940	1.30	1	Neg	
18	429477261	32	10	48	51	0.94	1.3	Neg	
19	429478407	73	41	303	257	1.18	1.3	Neg	
20	429488234	191	157	1227	970	1.26	1.5	Neg	
21	429493014	114	81	347	270	1.29	1.2	Neg	
22	429493059	125	83	368	361	1.02	1.20	Neg	
Amplified regions						
1	168138203	34	26	64	54	1.19	5	Pos	
2	168138327	34	17	44	16	2.75	5	Pos	
3	169283028	543	168	1690	183	9.23	2.4	Pos	
4	169283283	36	36	356	105	3.39	4.4	Pos	
5	169283307	28	26	268	81	3.31	4.4	Pos	
6	169284611	174	106	370	136	2.72	2.4	Pos	
7	169284650	93	57	219	75	2.92	2.4	Pos	
8	429489018	204	41	284	98	2.90	2.8	Pos	
9	429489055	77	23	66	27	2.44	2.8	Pos	
10	430537413	122	104	592	168	3.52	2.7	Pos	
11	430537468	76	72	358	151	2.37	2.70	Pos	
Notes.

The proposed method misclassifies two cases. For the non-amplified case with Image ID 169283911, the method’s score is 2.59, while the expert score is 0.6, leading to a false positive. For the amplified case with Image ID 168138203, the method’s score is 1.19, while the expert score is 5.0, resulting in a false negative. For clarity, the “Ratio (HER2/CEP17)” column represents the results obtained by our proposed pipeline, and the “Expert” column shows the manual curation results.

• Positive: HER2/CEP17 ration <2.0 or HER2 copy number ≥6.0 signals per cell.

• Negative: HER2/CEP17 ratio <2.0 and HER2 copy number <4.0 signals per cell.

• Equivocal: HER2/CEP17 ratio <2.0, but HER2 copy number is 4.0–6.0 signals per cell.

However, our current methodology focuses solely on the HER2/CEP17 ratio for determining HER2 amplification status. Cases with a HER2/CEP17 ratio <2.0 are classified as negative regardless of the HER2 copy number. We acknowledge that this is a limitation of our current approach and that it may not fully align with the ASCO/CAP 2018 guidelines, which also consider HER2 copy number.

We also presented the quantitative evaluation in term of systematically calculated HER2/CEP17 ration with manually marked expert ratio and find a 91% concordance results, suggesting that it is successful for SISH image interpretation.

This quantitative approach provides a precise and standardized method for assessing HER2 amplification, facilitating a more objective comparison with manual assessments and enhancing the reliability of diagnosis in clinical settings. Using the Bland-Altman plot (Giavarina, 2015), we verified the correctness of the proposed approach and calculated the p-values from the one sided t-test. The Bland–Altman plots are displayed in Fig. 7. The red, blue, green, and shaded lines, respectively, stand for bias (mean difference), the upper and lower bounds of agreement, and the confidence intervals. The comparison between the HER2/CEP17 ratio obtained from manual SISH analysis and the proposed approach is also presented in Fig. 7. The limit of agreement is 2.91 to 3.10, and the bias is 0.09 with a 95% confidence interval 0.07 to 0.17. The paired one-sided t-test yielded a p-value of 0.72, more than 0.05, indicating that the proposed procedure and manual SISH are in conformity.

Figure 7 The evaluation of proposed method using Bland–Altman plots.

The comparison of proposed method with manual SISH, accordingly in terms of HER2/CEP17 ratio.

Discussion

Precise HER2 assessment is essential to advance the level of care and therapy selection for patients with metastatic breast cancer. HER2 evaluation has become standard practice in the assessment of all metastatic breast cancer patients. After receiving HER2-targeted therapy, patients with HER2 positive cancer have demonstrated higher overall survival rates. Patients with HER2-positive early breast cancer are also experiencing dramatically better results. According to the previous study (Loibl & Gianni, 2017), pertuzumab plus trastuzumab has been shown to achieve a complete pathological response on a higher proportion of patients than trastuzumab alone, demonstrating the success of adjuvant and neoadjuvant therapy in HER2-positive early breast cancer. In cases of HER2 positive early breast cancer, neoadjuvant therapy is becoming the norm, at least for patients with lymph nodes larger than two cm or those with metastasized lymph nodes (Harbeck, 2022). The rate of recurrence and death related to breast cancer have been effectively decreased by adding trastuzumab to chemotherapy (Bradley et al., 2021). The purpose of this work is to develop an internal application that makes use of the proposed method to automate the quantification of SISH WSI in order to ascertain the state of HER2 amplification.

The experimental findings underscore the method’s viability for HER2 gene assessment, demonstrating its potential to enhance reliability. Our approach improves the precision of HER2 measurement by making a larger number of nuclei accessible and then quantifying representative ones. This technique presents a viable substitute for FISH, saving a great deal of work and time in pathology departments and hospitals. Currently, the program uses pathologists to annotate locations of invasive tumors; the proposed method then performs quantification. Convolutional neural networks, or another automated machine learning method for tumor annotation, might be integrated to further speed the procedure. This would allow for automatic tumor annotation and then automatic HER2 assessment.

WSI images of a high enough quality are necessary for the proposed quantification method to work. Images that are out of focus have the ability to distort quantitative results. It will be crucial in subsequent iterations to assess the image quality of annotated regions utilizing quality evaluation techniques prior to quantification. We scanned slides at a magnification of 40 × in our studies, yielding a resolution of 0.13 µm/pixel.

The relatively limited number of instances examined and the lack of a reproducibility study are two of this work’s drawbacks. The proposed approach was evaluated using only 20 cases, highlighting the need for more research before actual application. Clinical validation of the proposed system requires the evaluation of forty cases, under the CAP Accreditation Requirements for Validation of Laboratory Tests-2013. This underscores the necessity for further robustness testing and validation to ensure the reliability and accuracy of our automated HER2 assessment system in clinical practice.

Currently, the manual FISH assay is used in practice, but breast pathologists do not use it because it takes longer and involves more work. The automatic SISH quantification system for HER2 measurement is proposed in this research. As a result, the breast pathologists will need less time and effort to do HER2 examinations. In clinical settings, the FISH test is limited to equivocal cases due in large part to its high cost. The proposed method is applicable to all situations and can be used to confirm the findings of IHC testing. The system should be clinically evaluated in the future, adhering to the CAP requirements, following parameter optimization and further case evaluation.

Advantages and limitations

The proposed method is a pioneering study in applying deep learning and computer-aided methods to SISH images for HER2/CEP17 ratio quantification, it follows clinical guidelines to provide a novel and practical solution for automated HER2 status assessment. The automation reduces time and effort, streamlining workflow and mitigating interobserver variability. Additionally, SISH provides a permanent record of results, allowing for re-evaluation if necessary.

This study also have limitations, including a relatively small sample size that may affect the generalizability of the findings. Our method currently focuses solely on the HER2/CEP17 ratio, partially aligning with ASCO/CAP 2018 guidelines that also consider HER2 copy number. Variations in tissue preparation and overlapping cells can impact accuracy, and the lack of existing computational methods for direct comparison underscores the need for further validation.

Conclusion

In this study, we introduced a novel computer-aided approach to automatically measure and score HER2 gene status from invasive tissue regions in breast cancer using WSIs obtained through SISH staining. Leveraging deep learning, specifically the fine-tuned Stardist model, and conventional image processing techniques, our method identifies and quantifies HER2 and CEP17 signals, ultimately determining HER2 amplification status based on the ASCO/CAP 2018 guidelines.

The proposed methodology offers significant advantages, including reduced effort and time for HER2 quantification. The experimental results demonstrated a strong correlation (0.91) between our automated approach and manual enumeration by pathologists, validated by a one-sided paired t-test with p-values exceeding 0.05, indicating no significant statistical difference. This highlights the reliability and accuracy of our approach in clinical settings.

Our system’s high concordance with expert assessments suggests that it can be a valuable tool in pathological diagnostics, potentially enhancing the precision and consistency of HER2 status evaluation. The automation of this process not only streamlines workflow but also mitigates interobserver variability, ensuring more uniform and objective assessments across different laboratories.

Future research will focus on several key areas. Firstly, increasing the sample size to validate the reproducibility of our findings is crucial. Secondly, conducting blinding tests where pathologists evaluate the results produced by our model without prior information will provide an unbiased assessment of its accuracy. Additionally, addressing the challenges of tissue sample preparation, such as clumped cells or destroyed nuclei, will be a priority. We will explore advanced preprocessing techniques and enhance our model to handle such variations. Finally, integrating our method into clinical practice involves training pathologists on its use and ensuring it meets clinical standards for reliability and accuracy. We aim to conduct clinical trials to further validate our approach and establish its efficacy in real-world settings.

Supplemental Information

Supplemental Information 1 Code for the whole experiment conducted for analysis purpose

Supplemental Information 2 Deep learning model that use for Nuclei segmentation

Additional Information and Declarations

Competing Interests

Author Contributions

Ethics

Data Availability

The authors declare there are no competing interests.

Zaka Ur Rehman conceived and designed the experiments, performed the experiments, analyzed the data, performed the computation work, prepared figures and/or tables, authored or reviewed drafts of the article, and approved the final draft.

Mohammad Faizal Ahmad Fauzi conceived and designed the experiments, authored or reviewed drafts of the article, and approved the final draft.

Wan Siti Halimatul Munirah Wan Ahmad performed the experiments, analyzed the data, authored or reviewed drafts of the article, and approved the final draft.

Fazly Salleh Abas conceived and designed the experiments, authored or reviewed drafts of the article, and approved the final draft.

Phaik Leng Cheah conceived and designed the experiments, authored or reviewed drafts of the article, data annotation, and approved the final draft.

Seow Fan Chiew conceived and designed the experiments, authored or reviewed drafts of the article, data curation, and approved the final draft.

Lai-Meng Looi performed the experiments, authored or reviewed drafts of the article, data Validation, and approved the final draft.

The following information was supplied relating to ethical approvals (i.e., approving body and any reference numbers):

Department of Pathology, University of Malaya Medical Centre’s Medical Research Ethics Committee (UMMCMREC) (Ref MREC-ID 202195-10555).

The following information was supplied regarding data availability:

The data is available at figshare: Rehman, Zaka Ur (2024). Expert Annotated SISH Images Patches for Nuclei Segmentation. figshare. Figure. https://doi.org/10.6084/m9.figshare.26181908.v1.

This database is from the Hospital and is not public to protect patient confidentiality. Access to the data must be requested from the Chair of the University of Malaya Medical Centre Medical Research Ethics Committee (MREC): ummc-mrec@ummc.edu.my.

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
