# Peer review of "Computational approach for counting of SISH amplification signals for HER2 status assessment"

_PeerJ Computer Science, doi:10.7717/peerj-cs.2373_

## Round 0.1 · original submission · Major Revisions

The reviewers have substantial concerns about this manuscript. The authors should provide point-to-point responses to address all the concerns and provide a revised manuscript with the revised parts being marked in different color.

Reviewer 1 ·

Basic reporting

1. Introduction and Background:

Clarity and Detail: The introduction provides a general overview of HER2 quantification methods but lacks detailed background on SISH and its advantages over other methods such as FISH. Expand the introduction to include more context on SISH and the specific challenges it addresses.
Literature Review: The literature review is insufficient. There are references to key studies, but the review should be more comprehensive. Include recent studies and advancements in deep learning applications for pathology and HER2 quantification. Please add more background information about deep learning.

2. Methodology:

Detail and Reproducibility: The methods section lacks sufficient detail for reproducibility. Provide a step-by-step description of the deep learning model training process, including data preprocessing, augmentation techniques, hyperparameter tuning, and validation methods.
Justification of Methods: Explain why the chosen methods (e.g., Stardist model) are appropriate for this study. Include comparisons with other potential models or techniques that were considered.
Evaluation Metrics: Clearly define the metrics used to evaluate the model’s performance. Include precision, recall, F1-score, and any other relevant metrics.
3. Results:


Comparison with Existing Methods: Provide a detailed comparison of the proposed method’s performance against existing methods. Highlight the advantages and limitations in a separate subsection.
4. Discussion:

Depth of Analysis: The discussion should be expanded to provide a deeper analysis of the results. Discuss the implications of the findings, potential limitations of the study, and how these limitations could be addressed in future research.
Clinical Relevance: Emphasize the clinical relevance of the automated SISH quantification method. Discuss how this method could be integrated into clinical practice and its potential impact on patient outcomes.
Future Directions: Outline clear future directions for research based on the findings. Suggest specific studies or experiments that could build on this work.

Summary
The manuscript presents an innovative approach to automated HER2 quantification using SISH images and deep learning models. While the study has potential, substantial revisions are required to improve the introduction, methodology, results, discussion, and overall presentation. Addressing these issues will enhance the manuscript’s clarity, reproducibility, and impact.

Experimental design

2. Methodology:

Detail and Reproducibility: The methods section lacks sufficient detail for reproducibility. Provide a step-by-step description of the deep learning model training process, including data preprocessing, augmentation techniques, hyperparameter tuning, and validation methods.
Justification of Methods: Explain why the chosen methods (e.g., Stardist model) are appropriate for this study. Include comparisons with other potential models or techniques that were considered.
Evaluation Metrics: Clearly define the metrics used to evaluate the model’s performance. Include precision, recall, F1-score, and any other relevant metrics.

Validity of the findings

3. Results:


Comparison with Existing Methods: Provide a detailed comparison of the proposed method’s performance against existing methods. Highlight the advantages and limitations in a separate subsection.

Reviewer 2 ·

Basic reporting

1. The manuscript could benefit from professional editing services to polish the writing and eliminate typos and grammatical errors. For example, in line 144, “contract” should be “contrast”; in the Figure 5 legend, “market” should be “marked”.

2. All images lack scale bars.

3. In the title of the paper, the authors state that they are quantifying the HER2 ratio. I recommend changing this to HER2/CEP17 ratio or HER2 amplification status for accuracy.

4. Some of the figure titles start with “This is basically…”. I recommend deleting this part and using the rest as the figure titles.

5. Line 201: Does the “650,570 patches” mean 650 + 570 = 1220 patches?

6. Line 223: Please clarify what “high positivity” means.

Experimental design

Based on the 2018 ASCO/CAP standards, the HER2 amplification status is characterized as follows: Positive: HER2/CEP17 ratio ≥ 2.0 or HER2 copy number ≥ 6.0 signals per cell.
Negative: HER2/CEP17 ratio < 2.0 and HER2 copy number < 4.0 signals per cell.
Equivocal: HER2/CEP17 ratio < 2.0, but HER2 copy number is 4.0-6.0 signals per cell.
Using these standards, if the HER2/CEP17 ratio remains < 2.0 with ≥ 6.0 HER2 signals per cell, the diagnosis is HER2-positive. However, the pipeline described in this work characterizes this case as HER2-negative. Please revise the pipeline so that the diagnosis fully aligns with the 2018 ASCO/CAP standards.

Validity of the findings

1. The authors compared the results generated from their pipeline to the manual SISH quantification conducted by experts. The status listed in the last column reflects the results from manual curation but not the pipeline. I recommend the authors reorganize the results into two columns. Additionally, in some regions, such as non-amplified #10 and amplified #1, the results differ between the pipeline and the manual curation. It may be worth revisiting these cases to determine which result aligns more accurately with the actual amplification status.

2. All the subpanels in Figure 6 are not marked, making it hard to interpret the data. Furthermore, the authors concluded that the proposed technique for signal detection demonstrates superior visual performance compared to human observers, as shown in Figure 6(g–i). As this conclusion is not obvious from the provided figures, I recommend the authors add more details about how this conclusion was reached.

Additional comments

N.A.

Reviewer 3 ·

Basic reporting

no comment

Experimental design

no comment

Validity of the findings

no comment

Additional comments

HER2 gene is an important biomarker, and there are many ways to quantify it. Continuous deep learning models have made great progress in making quantification time faster and more accurate, and a strong correlation between continuous deep learning models and manual enumeration by pathologists provides evidence that the method can actually be reliable and accurate in clinical practice. However, there are still areas to be improved. As mentioned in the text, the sample size was too small to show reproducibility. It would be great if you could increase the sample number further. And if someone who has no knowledge of this experiment other than counting it manually by the pathologist (blinding test), it would be more helpful to evaluate whether the model is accurate. Finally, tissue samples are not always perfectly prepared, and it can be difficult to obtain data and reliable data values, especially if multiple cells are clumped together or the nucleus is severely destroyed. If there is a solution to this, please let me know.

---

## Round 0.2 · accepted · Accept

Reviewers are satisfied with the revisions, and I concur to recommend accepting this manuscript.

Reviewer 1 ·

Basic reporting

Authors have addressed all points from my review report. Strongly suggest for acceptance, thank you.

Experimental design

Authors have addressed all points from my review report. Strongly suggest for acceptance, thank you.

Validity of the findings

Authors have addressed all points from my review report. Strongly suggest for acceptance, thank you.

Additional comments

Authors have addressed all points from my review report. Strongly suggest for acceptance, thank you.

Reviewer 3 ·

Basic reporting

The HER2 gene is an important biomarker, and there are several ways to quantify it, this paper discusses the advantages of these methods compared to FISH. More comparisons than ever before are attached to show the advantages and need for these methods.

Experimental design

A computer-aided method is used for HER2 gene scoring for image processing and deep learning. This method is new and does not have many references. This paper explains the whole process and more information.

Validity of the findings

The specificity and advantages of this method over other methods are well described, and the authors also acknowledge a number of limitations of the study.

Additional comments

This version is a significant improvement over the previous version. The authors believe that this version addresses the points made by the three reviewers, clearly explains what the paper is trying to say shows how it does so, and recognizes the limitations of the current method and will address them in future work.